# Elevated Photovoltaic Performance in Medium Bandgap Copolymers Composed of Indacenodi-thieno[3,2-*b*]thiophene and Benzothiadiazole Subunits by Modulating the π-Bridge

**DOI:** 10.3390/polym12020368

**Published:** 2020-02-07

**Authors:** Lili An, Junfeng Tong, Yubo Huang, Zezhou Liang, Jianfeng Li, Chunyan Yang, Xunchang Wang

**Affiliations:** 1Key Laboratory for Utility of Environment-Friendly Composite Materials and Biomass in University of Gansu Province, School of Chemical Engineering, Northwest Minzu University, Lanzhou 730030, China; 2School of Materials Science and Engineering, Lanzhou Jiaotong University, Lanzhou 730070, China; yubohuang94@163.com (Y.H.); zezhouliang@hotmail.com (Z.L.); ljfpyc@163.com (J.L.); YChY-5@126.com (C.Y.); 3CAS Key Laboratory of Bio-Based Materials, Qingdao Institute of Bioenergy and Bioprocess Technology, Chinese Academy of Sciences, Qingdao 266101, China; wang_xc@qibebt.ac.cn

**Keywords:** indacenodithieno[3,2-*b*]thiophene, random conjugated polymer, modulating π-bridge, photovoltaic property

## Abstract

Two random conjugated polymers (CPs), namely, PIDTT-TBT and PIDTT-TFBT, in which indacenodithieno[3,2-*b*]thiophene (IDTT), 3-octylthiophene, and benzothiadiazole (BT) were in turn utilized as electron-donor (D), π-bridge, and electron-acceptor (A) units, were synthesized to comprehensively analyze the impact of reducing thiophene π-bridge and further fluorination on photostability and photovoltaic performance. Meanwhile, the control polymer PIDTT-DTBT with alternating structure was also prepared for comparison. The broadened and enhanced absorption, down-shifted highest occupied molecular orbital energy level (*E*_HOMO_), more planar molecular geometry thus enhanced the aggregation in the film state, but insignificant impact on aggregation in solution and photostability were found after both reducing thiophene π-bridge in PIDTT-TBT and further fluorination in PIDTT-TFBT. Consequently, PIDTT-TBT-based device showed 185% increased PCE of 5.84% profited by synergistically elevated *V*_OC_, *J*_SC_, and *FF* than those of its counterpart PIDTT-DTBT, and this improvement was chiefly ascribed to the improved absorption, deepened *E*_HOMO_, raised *μ*_h_ and more balanced *μ*_h_/*μ*_e_, and optimized morphology of photoactive layer. However, the dropped PCE was observed after further fluorination in PIDTT-TFBT, which was mainly restricted by undesired morphology for photoactive layer as a result of strong aggregation even if in the condition of the upshifted *V*_OC_. Our preliminary results can demonstrate that modulating the π-bridge in polymer backbone was an effective method with the aim to enhance the performance for solar cell.

## 1. Introduction

By virtue of the intriguing features of environmentally friendly, inexhaustibility, and widespread distribution, solar energy as one kind of green renewable energy has yielded more and more attention [1,2]. Among the multitudinous solar energy utilization technologies, bulk-heterojunction (BHJ) polymer solar cells (PSCs) whose photoactive layer consists of *p*-type π-conjugated polymers (CPs) and *n*-type small molecules (i.e., PCBM or ITIC etc.,) have received an increasing attention because of excellent superiorities, such as, low-cost, light weight, mechanical flexibility, and low temperature solution processing [3,4,5,6,7]. To date, power conversion efficiencies (PCEs) of PSCs have surpassed over 11% [8] in fullerene-accepter system and 15% in non-fullerene system [9], respectively, profited by the design of novel materials, optimization of morphology for photoactive layer and the deepened understanding of the structure-performance relationship [4,10,11,12,13,14,15,16,17,18]. In the past decades, tremendous efforts have been devoted to design and explore donor-acceptor (D-A) type CPs since their absorption, energy levels, molecular planarity, and thus charge mobility and morphology could be finely regulated by means of judiciously selecting the D and A subunits and alternating them into the polymer backbone so as to tune the photo-induced intramolecular charge transfer (ICT) effect [19,20,21,22,23,24]. With the benefits of the designing strategy, many excellent D-A copolymers involving PBDT-DTNT [25], PNTz4T-2OD [4], PDTBDT-BT [26], PM6 [27], PBDB-T-SF [28], and J61 [29] etc., have been developed. By comparison with the fully developed low band gap (LBG, *E*_g_ < 1.6 eV) copolymers, medium band gap (MBG, 1.6 < *E*_g_ < 1.8 eV) and/or wide band gap (WBG, *E*_g_ > 1.8 eV) copolymers were relatively less of concern [30,31]. However, MBG and WBG copolymers were still of importance in the PSCs, because the relatively large band gap cannot only render the low-lying highest occupied molecular orbital energy level (*E*_HOMO_) in order to achieve the higher open-circuit voltage (*V*_OC_), but also couple with the electron-acceptor so as to broaden the absorption range and to garner the high short-circuit current density (*J*_SC_) [32,33,34]. Consequently, it was of significant importance and highly essential to explore the new MBG and WBG copolymers and make a thorough inquiry of the structure-performance relationship.

Recently, highly coplanar aromatic/heteroaromatic units have attracted tremendous interest, because they possessed the following characteristics: (i) the elongated effective conjugation and improved overlap of π-orbitals along the polymer backbone direction in order to facilitate electron delocalization, which was instrumental in gaining a red-shifted absorption spectra to capture sunlight; (ii) the better intermolecular charge carrier hopping and restrained the interannular rotation so as to reduce the Marcus reorganization energy that could affect the rate of charge hopping, leading to an improved intrinsic charge mobility [22,24,35,36,37]. One representative ladder-type penta-ring indacenodithiophene (IDT) unit, by covalently fusing the outer thiophene rings into the central benzene core, has gained tremendous interest because of the following merits: (i) better coplanarity, highly electron density originated from large fused system, and desirable mobility; (ii) the bridge carbon (sp^3^ hybridization) which can be easily replaced by the heteroatoms such as Si, Ge, N, and S; (iii) multiple sites which can bond different side chain to ensure good solubility and solution-processability [37,38,39,40]. Therefore, substantial IDT-based and IDT-extended hepta-ring indacenodithieno[3,2-*b*]thiophene (IDTT)-based CPs have been developed [41,42,43,44,45,46,47,48,49]. The PCE was increased from 5.97 to 7.03% when a penta-ring IDT was replaced with an extending π-conjugation IDTT along the linear backbone [41]. Moreover, IDTT-T1 exhibited increased carrier mobility and more balanced carrier transport than that of IDT-T1 and thus the PCE was optimized from 6.26 to 6.58% [48]. When IDTT was further expanded to an undeca-ring IDT-biscyclopenta[2,1-*b*:3,4-*b*′]dithiophene (IDTCPDT), in which the central IDT core was fused by bilateral CPDT units, the synthesized fully rigidified PIDTCPDT-DFBT exhibited more wider and stronger absorption band, decreased reorganizational energy from 4.1 to 3.2 kcal mol^−1^, and 15-fold-elevated hole mobility and thus the 40% increased *J*_SC_ [36]. In addition to selecting the appropriate electron-rich D and electron-deficient A units to construct the efficient D-π-A type CPs, it has been proved that the conjugated π-bridges, usually consisting of conjugated subunits with small volume (i.e., thiophene (T), 2,2′-bithiophene (2T), thieno[3,2-*b*]thiophene (T), furan, selenophene (SeT), thiazole (Tz) and so on), played the vital role in constructing the highly efficient copolymers since these conjugated π-bridges not only linked the D and A moieties in the CPs within the D-π-A type structure, but also hindered the D and A units [38,43,50,51,52,53,54,55,56,57,58,59,60,61,62,63,64,65,66]. By inserting the flanked thiophene bridge on the two sides of thiadiazole[3,4-*c*]pyridine (PyT) or dibenzothiophene-*S*,*S*-dioxide units, the elevated *E*_HOMO_s never evidently down-shifted the *V*_OC_ and thus the increased PCE were achieved due to an improved *FF* [38,67]. For naohtho[1,2-*c*:5,6-*c*′]bis[1,2,3]triazole (TZNT) system, the adding of thiophene bridge enabled the red-shifted absorption and in turn a slightly increased *J*_SC_, however the rising *E*_HOMO_ severely limited the *V*_OC_, leading to a deteriorated PCE [58]. Extending the π-bridge from thiophene to TT, multiple investigation systems consisting of benzothiadiazole (BT) [51], difluoro-substituted-BT [43], dialkyloxy-substituted BT [54,64], thieno[3,4-*c*]pyrrole-4,6-dione (TPD) [56], thiazolo[5,4-*d*]thiazole (TzTz) [62] and so on were systematically investigated and the optimized PCEs were achieved. In parallel, no remarkable improvement of PCE in BDT-*alt*-fluorinated quinoxaline series [53] and even a decreased PCE in IDTT-*alt*-TzTz system [42] were observed after replacing thiophene with an enlarged TT. When substituting of thiophene with 2T, the improvement of PCE from 1.55% to 3.34% for TT-*alt*-BT system and one from 2.97% to 5.07% for IDT-*alt*-naphtho[1,2-*c*:5,6-*c*′]bis(1,2,5-thiadiazole) system were achieved principally benefited from the broadened absorption and optimized morphology of the photoactive layer [44,57]. The π-bridge effect of furan, thiophene, and TT on PV performance in BDT-*alt*-BT-based copolymers was investigated, and the gradually upshifted PCE from 2.81% to 3.72% to 4.93% profited from broadened absorption and elevated hole mobility and thus increased *J*_SC_ and *FF* in spite of a decreased *V*_OC_ as a result of the raised *E*_HOMO_ was observed [50]. In IDTT-*alt*-benzo[1,2-*c*:4,5-*c*′]dithiophene-4,8-dione (BDD)-based copolymer system, the thiophene bridge was substituted by selenophene and thiazole, and the PCE first increased then decreased from 7.04% to 8.65% then to 2.75% [65]. Interestingly, the thiazole bridge orientation in PTBTz-2 produced a widened absorption, deepened *E*_HOMO_, more planar configuration and thus 4.14-times higher PCE than that of PTBTz-5 [61]. Recently, our group cut down the thiophene bridge to prepare two random CPs PDTBDT-SBT and PDTBDT-SFBT, and found that there existed the almost unchanged band gaps in films, gradually increased *E*_HOMO_ levels, enhanced π–π stacking interaction, and the first decreased then increased photostability, when the number of the 3-octylthiophene bridge in each repeat unit varied from 0 to 1 and then up to 2 [18]. Besides, fluorination has also been proven to be a facile and effective strategy aiming at designing efficient CPs, because the energy level can be effectively descended without impairing bandgap as the result of the most electronegative nature of F, and relatively small volume can reduce the undesirable steric hindrance, as well as the molecular ordering was promoted by abundant hydrogen bonding interaction so as to speed up the exciton dissociation and prolongate the lifetime of charge carrier by reducing Coulombic interaction assisted by the induced dipole [68,69,70,71,72,73]. It should be noted that the fluorination can improve the molecular planarity and enhance the charge mobility but would inevitably reduce CPs’ solubility, resulting in the difficulty in the process of solution-processed fabrication [74]. These abovementioned results suggested that tuning the conjugated π-bridges and introducing fluorine into the polymer backbone were the facile and wise strategies that could effectively modulate the absorption, energy level, coplanarity, aggregation and charge transport, morphology of the photoactive layer, and thus the photovoltaic performance. Therefore, it is of importance to further investigate the effect of π-bridge in IDTT-consisting conjugated copolymers on the photovoltaic performance and photo-stability and further attempt to probe into the structure-property interplay.

Inspired by the upper importance and characteristic of conjugated π-bridge and fluorination, herein, two random CPs, namely PIDTT-TBT and PIDTT-TFBT coupled between an enlarged planarity bistin IDTTSn and asymmetric dibromides 4-bromo-7-(5-bromo-4-octylthien-2-yl)benzo-thiadiazole (TBTBr_2_) and/or 4-bromo-7-(5-bromo-4-octylthien-2-yl)-5,6-difluorobenzothiadiazole (TFBTBr_2_) were designed (Scheme 1). For clear comparison, the reference copolymer PIDTT-DTBT alternating IDTT and symmetric 4,7-bis(4-octyl-thien-2-yl)benzothiadiazole (DTBT) unit was also prepared. The impact of reducing thiophene π-bridge and fluorination on their absorption spectra, energy levels, aggregation, charge mobility, morphology of the photoactive layer, and thus photostability was systemically and detailly investigated. The adopted methods could broaden and increase absorption, lower *E*_HOMO_, improve the molecular planarity, and thus enhance film aggregation, but have little impact on aggregation in CB and photostability. Consequently, further photovoltaic measurement suggested that 185% increased PCE from 2.05 to 5.84% was achieved when reducing the thiophene π-bridge in PIDTT-TBT. However, the worsen PCE was observed after further introducing fluorine in PIDTT-TFBT.

## 2. Materials and Methods

### 2.1. Characterization

^1^H NMR spectra, melting points test, C, H, and N elemental analyses, TGA analyses, molecular weights of polymer, UV-Vis absorption, electrochemical properties, and atomic force microscopy (AFM) and transmission electron microscopy (TEM) images were obtained according to our reported methods [75]. The photostability of polymers and thin film X-ray diffraction (XRD) were measured according to our reported methods [34].

### 2.2. Materials

All reagents were of reagent grade and purchased from Sigma-Aldrich (Shanghai) Co. (Shanghai, China), Acros (Belgium, USA), J&K scientific (Beijing, China) and TCI (Shanghai) Chemical Co (Shanghai, China) and used without further purification, unless otherwise noted. The interlayer material 3,3′-(1,3,8,10-tetraoxoanthra[2,1,9-*def*:6,5,10-*d*′*e*′*f*′]diisoquinoline-2,9(1*H*,3*H*,8*H*,10*H*)diyl)-bis(*N*,*N*-dimethylpropan-1-amine oxide) (PDINO) was bought from Derthon optoelectronic materials science technology Co LTD (Shenzhen, China). Bistin 2,8-bis(trimethyltin)-6,6,12,12-tetra(4-hexylphenyl)-indacenodithieno[3,2-*b*]thiophene (IDTTSn) was prepared according to our reported method [34]. Symmetric dibromide 4,7-*bis*(5-bromo-4-octylthien-2-yl)benzo[*c*][1,2,5]thiadiazole (DTBTBr_2_) and unsymmetric 4-bromo-7-(5-bromo-4-octylthien-2-yl)benzo[*c*][1,2,5]thiadiazole (TBTBr_2_), as well as fluorinated 4-bromo-7-(5-bromo-4-octylthien-2-yl)-5,6-difluorobenzo[*c*][1,2,5]thiadiazole (TFBTBr_2_) were synthesized and described in Appendix A.

### 2.3. Polymer Synthesis

The resultant copolymers were synthesized as the following procedures: carefully purified bistin IDTTSn and dibromo-monomer (DTBTBr_2_, TBTBr_2_ and TFBTBr_2_) were fully dissolved into 6 mL degassed dry toluene and 0.8 mL DMF in a 25 mL two-neck round-bottom flask under Ar. The mixture was bubbled with Ar for another 20 min to remove O_2_. Thereafter, Pd_2_(dba)_3_ (1.4 mg) and P(o-tolyl)_3_ (2.3 mg) were quickly added to the mixture in one portion and the solution was bubbled with Ar for additional 20 min. The mixture was then vigorously refluxed for 48 h under Ar, followed by the subsequent addition of 2-tri(butylstannyl)thiophene and 2-bromothiophene at an interval of 8 h in order to finish ending-capping. After additional reflux at 8 h, the mixture was poured into 300 mL MeOH. The precipitate was collected by filtration and the crude polymer was subjected to Soxhlet extraction successively with MeOH, acetone, hexane, and toluene. Finally, the toluene fraction was condensed to approximately 6 mL and precipitated into MeOH. The black solid was collected and completely dried under vacuum overnight to obtain the target material with a yield of 92~98%.

#### 2.3.1. Synthesis of Poly[6,6,12,12-tetra(4-hexylphenyl)indacenodithieno[3,2-*b*]thiophene-2,8-diyl-*alt*-4,7-bis-(4-octylthien-2-yl)benzo[*c*][1,2,5]thiadiazole-5,5′-diyl] (PIDTT-DTBT)

IDTTSn (123.4 mg, 0.092 mmol) and DTBTBr_2_ (65.9 mg, 0.092 mmol) were used to prepare the PIDTT-DTBT according to the general procedure illustrated above. A black solid possessing metallic cluster of about 139.4 mg as reference polymer **PIDTT-DTBT** was obtained with a yield of 98.0%. Number-average molecular weights (*M*_n_) = 32.4 kDa, PDI = 1.8. ^1^H NMR (600 MHz, CDCl_3_), δ (ppm), 8.58 (s, ArH), 7.98 (s, ArH), 7.78 (br, ArH), 7.53 (dd, ArH), 7.41 (s, ArH), 7.23 (m, ArH), 7.12 (m, ArH), 2.84 (br, CH_2_), 2.57 (m, CH_2_), 1.74 (br, CH_2_), 1.58 (m, CH_2_), 1.44 (br, CH_2_), 1.35−1.20 (m, CH_2_), 0.87 (t, CH_3_). Anal. Calcd for C_98_H_110_N_2_S_7_: C, 76.41%; H, 7.20%; N, 1.82%. Found, C, 76.21%; H, 7.10%; N, 1.95%.

#### 2.3.2. Synthesis of Poly[6,6,12,12-tetra(4-hexylphenyl)indacenodithieno[3,2-*b*]thiophene-2,8-diyl-*co*-7-(4-octylthien-2-yl)benzo[*c*][1,2,5]thiadiazole-4,5′-diyl] (PIDTT-TBT)

IDTTSn (128.0 mg, 0.095 mmol) and TBTBr_2_ (46.46 mg, 0.095 mmol). The random polymer **PIDTT-TBT** was collected as black solid possessing metallic luster (118.9 mg, yield: 93.0%). *M*_n_ = 31.2 kDa, PDI = 1.9. ^1^H NMR (600 MHz, CDCl_3_), *δ* (ppm), 7.98 (s, ArH), 7.82 (br, ArH), 7.53 (s, ArH), 7.41 (s, ArH), 7.23 (dd, ArH), 7.12 (dd, ArH), 2.84 (br, CH_2_), 2.56 (m, CH_2_), 1.74 (br, CH_2_), 1.58 (m, CH_2_), 1.43 (br, CH_2_), 1.35−1.20 (m, CH_2_), 0.86 (m, CH_3_). Anal. Calcd for C_86_H_92_N_2_S_6_: C, 76.74%; H, 6.89%; N, 2.08%. Found, C, 76.51%; H, 6.79%; N, 2.18%.

#### 2.3.3. Synthesis of Poly[6,6,12,12-tetra(4-hexylphenyl)indacenodithieno[3,2-*b*]thiophene-2,8-diyl-*co*-7-(4-octylthien-2-yl)-5,6-difluorobenzo[*c*][1,2,5]thiadiazole-4,5′-diyl] (PIDTT-TFBT)

IDTTSn (136.8 mg, 0.102 mmol) and TFBTBr_2_ (53.32 mg, 0.102 mmol). The random polymer **PIDTT-TFBT** was collected as black solid possessing metallic luster (130.0 mg, yield: 92.0%). *M*_n_ = 34.4 kDa, PDI = 2.1. ^1^H NMR (600 MHz, CDCl_3_), *δ* (ppm), 8.67 (s, ArH), 8.12 (br, ArH), 7.57 (d, ArH), 7.54 (dd, ArH), 7.45 (s, ArH), 7.23 (m, ArH), 7.12 (m, ArH), 2.86 (br, CH_2_), 2.57 (m, CH_2_), 1.73 (br, CH_2_), 1.60 (br, CH_2_), 1.43 (br, CH_2_), 1.35−1.20 (m, CH_2_), 0.86 (t, CH_3_). Anal. Calcd for C_86_H_90_F_2_N_2_S_6_: C, 74.74%; H, 6.56%; N, 2.03%. Found, C, 74.51%; H, 6.41%; N, 2.19%.

### 2.4. Fabrication and Characterization of PSCs

The fabrication process for the devices used for photovoltaic performance measurement of the resultant copolymers was prepared according to our reported reference [13].

### 2.5. Hole-Only Device Fabrication and Measurement

The hole mobility of the active layer was measured from the *J*–*V* curves obtained under dark current described in our reported reference using the steady state space-charge-limited-current (SCLC) method [24,76].

## 3. Results and Discussion

### 3.1. Molecular Design, Synthesis, and Characterization

Key monomer bistin IDTTSn was synthesized according to our reported work [34]. Unsymmetrical dibromide TBTBr_2_ and TFBTBr_2_ are described in Appendix A. These key intermediates and comonomers have been identified by ^1^H NMR, as depicted in Appendix A and elemental analyses. These studied random copolymers PIDTT-TBT and PIDTT-TFBT as well as the control polymer PIDTT-DTBT were prepared via typical Stille polymerization, as elucidated in Scheme 1 [77] and further purified in the light of the reported method [75]. In the end, the fraction dissolved in toluene was recovered by re-precipitating in MeOH and then dried under vacuum overnight so as to remove the residual solvents. Note that all polymers were obtained as black solids exhibiting metallic luster in the high yields of 92~98%. It was found that these prepared copolymers displayed enough solubility in the chlorinated solvents (i.e., chloroform, chlorobenzene (CB), and *o*-dichlorobenzene (oDCB)), satisfying the requirements of the solution-processed fabrication procedure.

What is exciting is that the ^1^H NMR signals placed in the aromatic regions with regard to these resultant copolymers were all clearly observed, as outlined in Appendix A. In detail, peaks in the range of 8.67−7.41 ppm were produced by aromatic hydrogen from main chain of IDTT and DTBT, TBT and TFBT, and ones placed at 7.23 ppm and 7.11 ppm can be ascribed to aromatic hydrogen of hexylphenyl group. While, the peaks located at 2.86–2.84 ppm was belonged to the signal of –CH_2_– directly linked to the thiophene bridge and ones placed at 2.58–2.56 ppm were originated from the signal of –CH_2_– directly linked to phenyl group. Furthermore, the peaks ranged from 1.74 to 1.24 ppm were ascribed to the signals of remainder –CH_2_– of flexible side chains, and ones at 0.86 ppm were assigned to the signals of terminal –CH_3_. Besides, the results obtained from elemental analysis were found to be consistent with the theoretical values. Gel permeation chromatography (GPC) measurement exhibited the values for *M*_n_ and PDIs to be 32.4 kDa and 1.8 for PIDTT-DTBT, 31.2 kDa and 1.9 for PIDTT-TBT, and 34.4 kDa and 2.1 for PIDTT-TFBT, respectively (Appendix A), suggesting that the influence of the molecular weights on the optoelectronic and photovoltaic properties could be neglected. Seen from Appendix A, the control polymer PIDTT-DTBT and two random copolymers PIDTT-TBT and PIDTT-TFBT exhibited the decomposed temperature (*T*_d_, 5% weight-loss temperature) of approximately 310, 346, and 366 °C, respectively. Obviously, random polymer PIDTT-TBT with asymmetric π-bridge and further fluorinated PIDTT-TFBT exhibited a gradually elevated *T*_d_, which was in line with the enhanced molecular planarity and ordering obtained from the latter XRD analyses and DFT calculation.

### 3.2. Optical Property

In order to investigate the impact of reducing thiophene π-bridge and further fluorination on absorption behavior, the normalized UV-Vis absorption spectra in CB solution with the concentration of approximately 10^−5^ mol L^−1^ and as solid films were examined. As can be seen from Figure 1 and Table 1, the studied copolymers exhibited two legible absorption peaks, i.e., one situated at 350~500 nm were assigned to the π–π* transition from the resultant polymer backbone, the other one located in the range of 500~750 nm were originated from ICT effect from electron-rich IDTT unit to electron-deficient BT/FBT moiety [34,41]. After reducing the conjugated octylthiophene π-bridge and further fluorination, the decreased and blue-shifted variation (from 442 to 431 then to 428 nm in CB and from 445 to 436 then to 432 nm in film) for absorption peak in the high energy region in both CB and solid film was found, while only red-shifted change in the CB but both red-shifted and enhanced alternation for peak in the low energy region were also observed, these phenomena were identical with other groups’ results [41,43]. Ongoing from solution to film state, the red-shifted values for the maximum absorption peaks (λ_max_) of 6 nm for PIDTT-DTBT, 8 nm for PIDTT-TBT, and 18 nm for PIDTT-TFBT were observed, respectively. In particular, the fluorinated PIDTT-TFBT exhibited a weak peak seated at 590 nm, implying an enhanced molecular aggregation and stacking interaction evoked by fluorination [24,25,68]. Importantly, the absorption coefficient in film state was enhanced after reducing thiophene π-bridge and further fluorination. These changes could preliminarily suggest that the studied CPs exhibited the gradually increasing aggregation ability according to the sequence of PIDTT-DTBT, PIDTT-TBT, and PIDTT-TFBT. Furthermore, the optical bandgaps ($E_{g}^{\mathrm{opt}}$) values of 1.79 eV for PIDTT-DTBT, 1.76 eV for PIDTT-TBT, and 1.77 nm for PIDTT-TFBT in films were estimated with the absorption edge ($\lambda_{\mathrm{onset}}^{\mathrm{film}}$) in terms of $E_{g}^{\mathrm{opt}}$ = 1240/$\lambda_{\mathrm{onset}}^{\mathrm{film}}$, suggesting that the $E_{g}^{\mathrm{opt}}$ exhibited first reduced then slightly increased tendency in the order of PIDTT-DTBT, PIDTT-TBT, and PIDTT-TFBT. Evidently, after the thiophene π-bridge was reduced and further fluorinated, the absorption and absorption coefficient were improved, which was conducive to acquiring the higher *J*_SC_.

To examine the difference of reducing π-bridge and further fluorination on aggregation in solution, the temperature-dependent absorption (TD-Abs) spectra of the resultant copolymers in CB solution (ca. 10^−5^ mol L^−1^) ranging from 105 to 25 °C with a 10 °C interval was tested [24,78]. As elucidated in Figure 2, the studied copolymers exhibited the similar variation tendency, that is, the ICT and π‒π* absorption peak were all blue-shifted and the absorption intensity decreased with the elevating temperature. In detail, after temperature was risen from 25 to 105 °C, the blue-shifted values (*Δ*λ) and the decreased absorption intensity (Δ*I*) were 14 nm (from 570 to 556 nm) and 15.1% for PIDTT-DTBT, 13 nm (from 602 to 589 nm) and 12.1% for PIDTT-TBT and 16 nm (from 602 to 586 nm) and 14.4% for PIDTT-TFBT, respectively. The observed similar vibration values with respect to absorption peak and intensity implied that the strategies of reducing π-bridge and fluorination produced the little impact on aggregation in CB solution [79].

### 3.3. Photostability Property

Aimed at achieving the industrialization of PSCs, in addition to the ability of harvesting sunlight, the photo-stability of organic semiconductor materials was also an important issue [18,34,80,81]. Consequently, in order to inspect the effect of reducing conjugated thiophene π-bridge and further fluorination on the photo-stabilities, the absorption variations of the resultant CPs were examined by exposing the diluted CB solution and the polymers’ films under AM1.5 sunlight irradiation in air at room temperature. As exhibited in Appendix A, we can see that after 4 h of photodegradation in CB solution, PIDTT-DTBT, PIDTT-TBT, and PIDTT-TFBT exhibited the almost unchanged absorption, that is to say, higher than 98% of their initial light absorption was maintained. As for the solid film, the UV-vis spectra changes of these copolymers are outlined in Figure 3. The peak related to π–π* transition almost remained unchanged and one assigned to ICT transition exhibited gradual descending with the time of light-soaking. In detail, 92.7%, 91.1%, and 93.9% of their initial light absorption for PIDTT-DTBT, PIDTT-TBT, and PIDTT-TFBT after 9 h of illumination were seen, which was obviously superior to that of PIDTT-DTNT-based copolymers [34]. Based on the above fact, it was concluded that the strategies of reducing thiophene π-bridge and further fluorination produced very tiny impact on the photostability in both CB and solid film.

### 3.4. X-ray Diffraction (XRD) Analysis

X-ray diffraction (XRD) analyses of the pristine polymers were applied to further probe into the influence of reducing π-bridge and further fluorination on crystallinity and molecular packing in solid film state, as disclosed in Figure 4. It was noted that the used films were casted from CB solution onto the glass substrate. There was only one weak diffraction peak at approximately 20.45° in the reference polymer PIDTT-DTBT, which was attributed to π–π stacking distance of 4.34 Å calculated from the Bragg’s law (i.e., λ = 2dsinθ) [34]. For PIDTT-TBT and PIDTT-TFBT, there existed two distinct diffractions. The sharp diffraction peaks in a small angle region are located at 2θ of 4.13° for PIDTT-TBT and 4.32° for PIDTT-TFBT, corresponding to the distance of polymer backbone separated by the flexible side were 21.37 and 21.01 Å, respectively. While the broad diffraction peaks placed at wide angle district located at 2θ of 20.55° for PIDTT-TBT and 21.98° for PIDTT-TFBT could be attributed to π–π stacking interaction, which reflected the corresponding π–π stacking distances of 4.32 and 4.04 Å, respectively. It is inescapably clear that the aggregation trend in the film state was enhanced when the thiophene π-bridge was reduced and further fluorinated, which agreed with the previous phenomenon from absorption.

### 3.5. Electrochemical Property

Since the *E*_HOMO_ and *E*_LUOMO_ values for photovoltaic materials were regarded as important parameters for estimating the photovoltaic performance, cyclic voltammetry (CV) method was utilized to gain insight into the impact of reducing the thiophene π-bridge and further fluorination on the electrochemical properties. It was noted that the *E*_HOMO_ and *E*_LUMO_ were calculated from its oxidation onset potential ($\varphi_{\mathrm{ox}}^{\mathrm{onset}}$) and reduction onset potential ($\varphi_{\mathrm{red}}^{\mathrm{onset}}$), respectively. As displayed in Figure 5a and Table 1, the corresponding $\varphi_{\mathrm{ox}}^{\mathrm{onset}}$ and $\varphi_{\mathrm{red}}^{\mathrm{onset}}$ values of 0.59 and –1.22 V for PIDTT-DTBT, 0.66 and –1.11 V for PIDTT-TFBT, and 0.73 and –1.04 V for PIDTT-TFBT were observed, respectively. It was needed to point out that the CV curve was recorded relative to the potential of the standard Ag/AgNO_3_ electrode, calibrated by the ferrocene-ferrocenium (Fc/Fc^+^) redox pair. The *φ*_1/2_ of the Fc/Fc^+^ redox pair was 0.11 V vs. Ag/AgNO_3_ electrode. Supposing that the redox potential for Fc/Fc^+^ was −4.80 eV in relative to vacuum energy level, the *E*_HOMO_ can be estimated from the formula *E*_HOMO_ = –e($\varphi_{\mathrm{ox}}^{\mathrm{onset}}$ + 4.69) (eV) and *E*_LUMO_ = –e($\varphi_{\mathrm{red}}^{\mathrm{onset}}$ + 4.69) (eV), respectively [82]. Consequently, the *E*_HOMO_ and *E*_LUMO_ values were estimated to be –5.28 and –3.47 eV for PIDTT-DTBT, –5.35 and –3.58 eV for PIDTT-TBT, and –5.43 and –3.65 eV for PIDTT-TFBT, respectively. Moreover, the electrochemical bandgap ($E_{g}^{\mathrm{ec}}$) values for PIDTT-DTBT, PIDTT-TBT, and PIDTT-TFBT were calculated to be 1.81, 1.77, and 1.77 eV, respectively. In order to make a better comparison, the schematic diagram with respect to the energy levels for the resulted copolymers and electron-acceptor PC_71_BM is depicted in Figure 5b. It was seen that the decreasing *E*_HOMO_ values ca. 0.07 and 0.15 eV were observed after reducing the thiophene π-bridge and further fluorination, respectively. This change trend was instrumental in yielding the higher *V*_OC_, since the *V*_OC_ was maximized with the enlarging difference between the *E*_HOMO_ of donor and *E*_LUMO_ of the acceptor [3,83]. Simultaneously, the differences between *E*_LUMO_ of donor polymers and one of *E*_LUMO_ of PC_71_BM ranged from 0.55 to 0.73 eV, which could support enough driving force to promote the exciton to dissociate into free charge at the D-A interfaces [3]. Theoretically, these decreased *E*_HOMO_ levels and the lowered bandgaps were hopeful for improving the *V*_OC_ and *J*_SC_, and thus resulting in the enhanced PCE.

### 3.6. Theoretical Calculation

The density functional theory (DFT) calculation situated at B3LYP/6-31G* basis set (Gaussian 09) was selected to further probe into the effect of reducing thiophene π-bridge and further fluorination on the molecular backbone conformation and the electron density distributions [84]. For simplification, the hexyl side chain from IDTT and octyl side chain attached onto the thiophene bridge were both replaced with the methyl, and oligomers involving one repeating unit (unimer) were used to take the place of the corresponding polymer backbones. Note that the unimer model for the studied random copolymers PIDTT-TBT and further fluorinated PIDTT-TFBT had two different linking patterns, (IDTT-BT-T) and/or (IDTT-T-BT), and (IDTT-FBT-T) and/or (IDTT-T-FBT), respectively. As presented in Figure 6, the HOMO orbitals of all studied compounds are primarily delocalized across the whole conjugated main chain, in contrast, the LUMO orbitals are preferentially distributed in electron-deficient BT and/or FBT and lateral thiophene bridges, indicating that there existed the relatively obvious charge-transfer effect from IDTT to BT and/or FBT units. It was seen that the theoretically forecasted values for *E*_HOMO_, *E*_LUMO_, and bandgap were −4.94, −2.80, and 2.14 eV of PIDTT-DTBT, −4.97, −2.79, and 2.18 eV of IDTT-BT-T, and −4.99, −2.69, and 2.30 eV of IDTT-BT-T for thiophene-bridge-reducing PIDTT-TBT, −5.04, −2.90, and 2.14 eV for PIDTT-FBT-T and −5.02, −2.86, and 2.16 eV of IDTT-T-FBT for further fluorinated PIDTT-TFBT, respectively. We can see that, regardless of the position of thiophene π-bridge, the descending trend of *E*_HOMO_ from PIDTT-DTBT to PIDTT-TBT then to PIDTT-TFBT was observed, which was in accordance with the results obtained from CV testing.

Since the good planarity in polymer backbone is helpful to promote closer π−π stacking, higher charge transfer, and enlarged molecular conjugation so as to generate the lower band gap for harvesting more sunlight [24,75]. Thus, we further inspected the dihedral angles *θ*_1_ between IDTT unit and alkylthiophene bridge, *θ*_2_ and *θ*_3_ between alkylthiophene bridge and BT and/or moiety. According to the data of Table 2, in spite of position of the thiophene π-bridge in the polymer backbone, the backbone planarity was all improved, meanwhile, the fluorination further optimized the backbone planarity. Definitely, these variations concerning energy level and molecular planarity can well agree with the results measured in former CV and XRD analyses.

### 3.7. Photovoltaic Properties

In order to probe into the impact of reducing thiophene π-bridge and further fluorination in IDTT-based polymer backbone on the photovoltaic properties, the devices with the configuration of ITO/PEDOT:PSS/polymers:PCBM/PDINO/Al were fabricated. Note that the photoactive layers were prepared by spin-coating the solution involving the studied copolymers and PC_61_BM and/or PC_71_BM, and the ultrathin interfacial layer (spinning from 1 mg L^−1^ PDINO solution in methanol) was utilized as cathode-modified interlayer [13]. The photovoltaic performance was optimized by the following processing, screening out the D/A ratio, utilizing the 1,8-diiodioctane (DIO) processing additive and substituting PC_61_BM with PC_71_BM. These device fabrication processes are listed in Appendix A in detail. After screening the D/A ratios, 1:1, 1:2 to 1:3, it was exhibited that the optimal D/A ratio for PIDTT-DTBT, PIDTT-TBT, and PIDTT-TFBT was 1:2, as shown in Appendix A. Seen from Appendix A, the control device based on PIDTT-DTBT exhibited the best PCE of 2.05%, with a *V*_OC_ of 0.81 V, a *J*_SC_ of 5.51 mA cm^−2^, and a *FF* of 45.67%, and PIDTT-TBT-based device showed an enhanced PCE of 4.48%, with the collaboratively enhanced *V*_OC_ of 0.87 V, *J*_SC_ of 10.44 mA cm^−2^ and *FF* of 49.30%; however, further fluorinated PIDTT-TFBT-based device showed an inferior PCE of 2.60%, with both decreased *J*_SC_ of 5.63 mA cm^−2^ and *FF* of 49.08% and a further elevated *V*_OC_ of 0.94 V. Meanwhile, the corresponding external quantum efficiency (EQE) curves corroborated the changes relative to *J*_SC_, as depicted in Appendix A.

Since processing solvent additive with high boiling point was conducive to forming more ordered and nano-scaled bicontinuous interpenetration network structure which can improve the dissociation of light-induced exciton into the free charge carriers at the D/A interface and accelerate the transport of free charge carriers [85,86]. Hence, following that, we selected the 3% DIO (DIO/CB, V/V) as a solvent additive to proceed to optimize the device performance, as illustrated in Appendix A. Unfortunately, a 14.6% deterioration of PCE (from 2.05 to 1.75%) in the control polymer PIDTT-DTBT-based device was found, mainly originating from the 18.2% decline in *FF* (from 45.67 to 437.37%) the even though the slight increases of *V*_OC_ (from 0.81 to 0.84 V) and *J*_SC_ (from 5.51 to 5.57 mA cm^−2^). As for PIDTT-TBT-based devices, it was exhibited that the 10.3% elevation in PCE (from 4.48 to 4.94%) was observed, chiefly benefiting from the 7.8% enhancement in *FF* (from 49.30 to 53.14 %) even if *V*_OC_ and *J*_SC_ kept relatively constant. A 12.3% increase in PCE (from 2.60 to 2.92%) was inspected in the further fluorination of PIDTT-TFBT-based device, primarily benefiting from slightly enhancements of 5.9% in *J*_SC_ (from 5.63 to 5.95 mA cm^−2^) and 5.0% in *FF* (49.08 to 51.54%) under the condition of the almost unchanged *V*_OC_. Apparently, positive effect in PIDTT-TBT and fluorinated PIDTT-TFBT but negative impact in the reference material PIDTT-DTBT on the photovoltaic performance were observed.

On account of PC_71_BM possessing relatively higher absorption coefficient and broader absorption band than those of PC_61_BM, herein PC_71_BM was applied to replace PC_61_BM for further enhancing the PCE [87], as displayed in Appendix A. As shown in Appendix A, the enhancement of 10.9% (from 1.75 to 1.84%) for PCE in PIDTT-DTBT-based device was achieved, primarily originating from 10.1% increase in *FF* (from 37.37 to 41.15%). Inspiringly, the remarkable 18.2% improvement of PCE for PIDTT-TBT was obtained, mainly benefiting from concurrently upshifted by 5.1% *J*_SC_ (from 10.54 to 11.08 mA cm^−2^) and 12.2% *FF* (from 53.14 to 559.60%) but the stable *V*_OC_. However, as for the fluorinated PIDTT-TFBT-based device, a 8.6% decline in PCE (from 2.92 to 2.67%) was obtained, mostly limited by 8.6% drop in *J*_SC_ (from 5.95 to 5.44 mA cm^−2^) but constant *V*_OC_ and almost unvaried *FF*. It was noted that these relevant *J*_SC_ variations were also confirmed by corresponding EQE curves in Appendix A.

With the assistance of D/A ratio screening, DIO additive and utilizing PC_71_BM to replace PC_61_BM, the optimal *J-V* characteristics and corresponding EQE spectra were elucidated in Figure 7. Seen from Table 3, for as much as the moderate electron-donating IDTT and strong electron-withdrawing BT subunits was selected to construct the polymer backbone, these yielded photovoltaic devices exhibited relatively higher *V*_OC_ ranging from 0.81 to 0.95 V. Meanwhile, the gradual increasing tendency of *V*_OC_ (from 0.81 to 0.88 then to 0.95 V) in the order of PIDTT-DTBT, PIDTT-TBT, and PIDTT-TFBT was in accordance with the above deepened *E*_HOMO_ predicted by DFT and CV tests [24]. We can see that the best PCE was remarkably increased from 2.05 to 5.84% after reducing the thiophene π-bridge, chiefly benefiting from collaborative 8.6% increase in *V*_OC_ (from 0.81 to 0.88 V), 101% enhancement in *J*_SC_ (from 5.51 to 11.08 mA cm^−2^), and 30.5% elevation in *FF* (from 45.67 to 59.60%). Conversely, 50% downshifted PCE (from 5.84 to 2.92%) was found after further fluorination, which was mainly limited by 46.4% decrease in *J*_SC_ (from 11.08 to 5.95 mA cm^−2^) and 13.52% drop in *FF* (from 59.60 to 51.54%) even if the *V*_OC_ increased by 18% (from 0.88 to 0.95 V). Theses *J*_SC_ variations were also verified by the corresponding EQE spectra (Figure 7b). Another point that needs to be emphasized is that the integrated *J*_SC_ values in terms of EQE curves were estimated to be 5.45, 11.00, and 5.80 mA cm^−2^ for the optimal PIDTT-DTBT:PC_61_BM, PIDTT-TBT:PC_71_BM, and PIDTT-TFBT:PC_61_BM, respectively, indicating there existed the tolerated error (<5%) with respect to *J*_SC_ values measured from *J-V* curves, demonstrating these *J*_SC_ values were reliable. Furthermore, the first enhanced then decreased change trend for *FF* was also confirmed by the first increased then decreased shunt resistance (*R*_SH_) (from 565 to 604 then to 536 Ω m^2^) and first decreased then increased series resistance (*R*_S_) (from 43.4 to 7.7 then to 14.9 Ω m^2^) when reducing the thiophene π-bridge and the further fluorination.

### 3.8. Charge Mobilities

In order to find the reason why reducing thiophene π-bridge and further fluorination in IDTT-based polymer backbone produced different impact on device performance, the vertical hole and electron transport properties were examined by hole-only and electron-only devices with the corresponding device configurations of ITO/PEDOT:PSS/blend/MoO_3_/Ag and ITO/ZnO/blend/MoO_3_/Al, respectively. The prepared photoactive layer films were fabricated under the same conditions with their best PSCs. These charge mobilities were calculated according to SCLC method which can be described by the equation $J=\;\frac{9}{8}\varepsilon_{0}\varepsilon_{r}\mu_{h}\frac{V^{2}}{L^{3}}$ [76]. Note that these thickness values of resultant photoactive layer blend films for PIDTT-DTBT, PIDTT-TBT, and PIDTT-TFBT were 98, 100, and 105 nm for hole-only devices and 115, 118, and 100 nm for electron-only devices (Appendix A), respectively. The *J-V* curves under dark are presented in Appendix A and the corresponding fitting *J*^1/2^−*V* curves for the photoactive layer films are shown in Figure 8. It was observed that the values for hole-mobility (*μ*_h_), electron-mobility (*μ*_e_), and *μ*_h_/*μ*_e_ ratio were 1.78 × 10^−4^, 8.56 × 10^−7^ cm^2^ V^−1^ s^−1^, and 2074 for PIDTT-DTBT, 7.67 × 10^−4^, 4.07 × 10^−7^ cm^2^ V^−1^ s^−1^, and 1884 for PIDTT-TBT, and 5.29 × 10^−4^, 1.77 × 10^−6^ cm^2^ V^−1^ s^−1^, and 299 for PIDTT-TBT (Table 4), respectively. Apparently, the hole-mobility showed a first increased then decreased trend, and the electron-mobility exhibited a gradually increased tendency as well as *μ*_h_/*μ*_e_ exhibited a decreased tendency after reducing the thiophene π-bridge in PIDTT-TBT and further fluorinating in PIDTT-TFBT compared to PIDTT-DTBT. These observed increased *μ*_h_, *μ*_e_ and more balanced *μ*_h_/*μ*_e_ values can in part account for the 30.5% increased *FF* and 101% elevated *J*_SC_ after reducing the thiophene π-bridge in PIDTT-TBT [88]. Moreover, the decreased *μ*_h_ after further fluorination in PIDTT-TFBT can explain the reason to some extent why the 13.52% deteriorated *FF* and 46.39% decreased *J*_SC_ were observed, which could be further supported by the latter oversized aggregation seen from AFM and TEM analyses.

### 3.9. Film Morphology

It is known to all that photovoltaic performance are strongly depended on the morphology of the photoactive layer in BHJ PSCs [21,86,89,90,91]. Consequently, in order to deeply make a thorough inquiry into the causes why reducing the thiophene π-bridge in PIDTT-TBT had a positive impact on PCE but the further fluorination had a negative effect on PCE in PIDTT-TFBT, the morphologies of the photoactive layer films prepared under exactly similar condition with those of optimal device were examined by AFM in a surface area of 5 × 5 μm. As seen from Figure 9, it was exhibited that the both PIDTT-DTBT and PIDTT-TBT-based blend films had very similar surface morphology with the nearest root-mean-square (RMS) roughness of 0.759 and 0.587 nm, respectively, and the fluorinated PIDTT-TFBT-based film displayed greatly increased RMS value of 2.071 nm; these vibrations as a whole were in accordance with the solid aggregation trend observed in the preceding XRD analysis. Furthermore, we utilized the TEM to further examine the in-depth morphology information of the photoactive layers, as depicted in Figure 10. We can see that when reducing thiophene π-bridge in PIDTT-TBT exhibited the slightly optimized phase separation compared with the control polymer PIDTT-DTBT, in the meantime, further fluorination in PIDTT-TFBT exhibited oversized aggregation restricting the effective exciton diffusion and decreasing the exciton dissociation into the free charges via reducing D/A interfacial areas as well as enhancing the charge recombination, which would lead to a detrimental *FF* and *J*_SC_ [21]. These morphology information can in part account for the improved photovoltaic performance in PIDTT-TBT-based device but deteriorated performance after further fluorination in PIDTT-TFBT were acquired.

## 4. Conclusions

To conclude, two IDTT-based random CPs, PIDTT-TBT and PIDTT-TFBT as well as the alternating copolymer PIDTT-DTBT as control material, were designed to study the influence of reducing thiophene π-bridge and further fluorination on photostability and photovoltaic performance. Because of selecting the moderate electron-donating IDTT and strong electron-withdrawing BT to construct the polymer backbone, the wide bandgap of 1.76~1.79 eV, lowlying *E*_HOMO_ ranging from −5.28 to −5.43 eV and excellent photostability were achieved. It was exhibited that the broadened and increased absorption, deepened *E*_HOMO_, improved molecular planarity, and thus enhanced film aggregation, but tiny impact on aggregation in CB solution and photostability were observed when PIDTT-DTBT was replaced with PIDTT-TBT and PIDTT-TFBT successively. The best photovoltaic tests disclosed that, PIDTT-TBT-based device yielded 185% increased PCE from 2.05 to 5.84% benefiting from simultaneously elevated *V*_OC_ of 0.88 V, *J*_SC_ of 11.08 mA cm^−2^, and *FF* of 59.60% compared to the counterpart, and this improvement was chiefly profited by the broadened and increased absorption, deepened *E*_HOMO_, raised *μ*_h_, and more balanced *μ*_h_/*μ*_e_, and slightly optimized morphology of photoactive layer. Conversely, the 50% reduced PCE was observed after further fluorination limited by the minimized *J*_SC_ and *FF*, mainly limited by undesired morphology for photoactive layer film as a result of strong aggregation despite of the enlarged *V*_OC_. This work suggested tuning the π-bridge in polymer backbone was an easy-to-implement and effectual tactic with a view to enhancing the photovoltaic property.

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
