# Peer review of "Elevated Photovoltaic Performance in Medium Bandgap Copolymers Composed of Indacenodi-thieno[3,2-b]thiophene and Benzothiadiazole Subunits by Modulating the π-Bridge"

_polymers, 2020, doi:10.3390/polym12020368_

Round 1
Reviewer 1 Report
The authors here report on the synthesis, characterization, and solar cell device performance of three donor-acceptor polymers. Overall, there is a thorough amount of data collected that provide a great deal of insight into these materials, and for the most part the data is carefully collected. The introduction is also well laid out, and clearly explains the motivation for the study.
There are a few areas of criticism:
In the section on absorbance spectra, there is a lot of discussion about the absorption coefficient of these materials. It is very unusual to report the absorption coefficient of polymers, because it is not possible to identify the chromophore, and therefore not possibly to calculate the moles of chromorphore. The chromophore could be made up of one repeat unit, or two repeat units, as well as mixtures of twisted, aggregated, and unaggregated units. This is true for both the values in solution, as well as in thin films. In Figure S10 the authors report on the absorption as a function of concentration, but this is just demonstrating Beer-Lambert Law, and is not scientifically interesting. I would remove all sections where absorption coefficient is discussed, and I would also remove Figures S10 and S11. I don't think they provide any insight into this study, and I also think the conclusions in this paper are strong even without these sections.
E-LUMO is calculated by first measuring the oxidation potential, converting this to E-HOMO, and subtracting the optical bandgap. If the authors have access to performing electrochemistry in a glovebox, it would be preferred if the reduction potential could be measured electrochemically. There are many problems with estimating E-LUMO by subtracting optical bandgap (see, for example, the manuscript Mind the Gap! by Jean-Luc Bredas in Materials Horizons, 2014 1, 17-19). However, if the authors do not have access to this setup, then it is understandable.
In Figure 4, the authors mention there is "no obvious diffraction peak in the reference polymer", however there is clearly a peak between 20-25 degrees that needs to be discussed.
In Figure 9, the AFM of image e) has some problems with the image: the lines that are visible in the scan suggest there was a problem with the scan itself. This image needs to be recollected. Figures d and f should be rescaled. It is more common to be in the +/- 5 or 10 degree range. Figure b is fine since it is showing some issues in the phase image.
In Figure 10, it is difficult to read the scale bar. Please either make the font larger in the image, or mention in the figure caption what the scale bar represents.
There are several places where the authors remark on unfortunate or frustrating results, such as in the abstract, or throughout the paper. It is not necessary to phrase things this way. The results can be pretty well explained from the data collected and well understood. There is nothing negative about this.
Finally, there is a significant amount of errors in the English that need to be corrected as it makes the paper very difficult to understand. Please take the time to do this, as it would very much help the quality of this manuscript.
Author Response
Reviewer #1:The authors here report on the synthesis, characterization, and solar cell device performance of three donor-acceptor polymers. Overall, there is a thorough amount of data collected that provide a great deal of insight into these materials, and for the most part the data is carefully collected. The introduction is also well laid out, and clearly explains the motivation for the study.
There are a few areas of criticism:
1) In the section on absorbance spectra, there is a lot of discussion about the absorption coefficient of these materials. It is very unusual to report the absorption coefficient of polymers, because it is not possible to identify the chromophore, and therefore not possibly to calculate the moles of chromorphore. The chromophore could be made up of one repeat unit, or two repeat units, as well as mixtures of twisted, aggregated, and unaggregated units. This is true for both the values in solution, as well as in thin films. In Figure S10 the authors report on the absorption as a function of concentration, but this is just demonstrating Beer-Lambert Law, and is not scientifically interesting. I would remove all sections where absorption coefficient is discussed, and I would also remove Figures S10 and S11. I don't think they provide any insight into this study, and I also think the conclusions in this paper are strong even without these sections.
Response: According to your advice, the description part related absorption coefficient in solution and solid films in the main text as well as the related Figures (Fig. S10 and Fig. S11) and Table (Table S2) in the supporting information have been omitted. The enhanced absorption coefficient description “Importantly, the absorption coefficient in film state was enhanced after reducing thiophene π-bridge and further fluorination.” has been added in the new manuscript.
2) E-LUMO is calculated by first measuring the oxidation potential, converting this to E-HOMO, and subtracting the optical bandgap. If the authors have access to performing electrochemistry in a glovebox, it would be preferred if the reduction potential could be measured electrochemically. There are many problems with estimating E-LUMO by subtracting optical bandgap (see, for example, the manuscript Mind the Gap! by Jean-Luc Bredas in Materials Horizons, 2014 1, 17-19). However, if the authors do not have access to this setup, then it is understandable.
Response: Thank you very much for your suggested reference. We have tried our best to carry out the CV reduction peak measurements under the nitrogen-bubbling conduction and the obtained results have been added in the revised manuscript. And the related description and Figure 5 and Table 1 have also been revised.
3) In Figure 4, the authors mention there is "no obvious diffraction peak in the reference polymer", however there is clearly a peak between 20-25 degrees that needs to be discussed.
Response: According to your advice, the discussion about the reference polymer PIDTT-DTBT has been added in the revised manuscript.
4) In Figure 9, the AFM of image e) has some problems with the image: the lines that are visible in the scan suggest there was a problem with the scan itself. This image needs to be recollected. Figures d and f should be rescaled. It is more common to be in the +/- 5 or 10 degree range. Figure b is fine since it is showing some issues in the phase image.
Response: According to your suggestions, the AFM Figures (Fig. 9e and 9f) have been checked and revised. And the Figure 9d was also rescaled.
5) In Figure 10, it is difficult to read the scale bar. Please either make the font larger in the image, or mention in the figure caption what the scale bar represents.
Response: According to the suggestion, the description of “The scale bar of TEM images is 100 nm.” has been added in the corresponding figure caption.
6) There are several places where the authors remark on unfortunate or frustrating results, such as in the abstract, or throughout the paper. It is not necessary to phrase things this way. The results can be pretty well explained from the data collected and well understood. There is nothing negative about this.
Response: The corresponding emotional descriptions have been revised according to the advices.
(1) In line 30 of Page 1, “Frustratingly” has been revised by “However”.
(2) In line 102 of Page 3, “frustrating” has been deleted.
(3) In line 147 of Page 4, “Disappointingly” has been revised by “However”.
(4) In line 440 of Page 12, “Regretfully” has been revised by “However”.
(5) In line 546 of Page 16, “Unfortunately” has been revised by “Conversely”.
7) Finally, there is a significant amount of errors in the English that need to be corrected as it makes the paper very difficult to understand. Please take the time to do this, as it would very much help the quality of this manuscript.
Response: Thank you very much for your kind suggestions. We have carefully checked the whole manuscript and corrected them.
For example,
(1) In line 34 of Page 1, “with the aim to enhancing” has been revised by “with the aim to enhance”.
(2) In line 73 of Page 2, “interesting” has been revised by “interests”.
(3) In line 86 of Page 2, a word “the” has been removed in front of “40%”.
(4) In line 91 of Page 2, “with” has been revised by “within”.
(5) In line 101 of Page 3, “Parallelly” has been revised by “In parallel” and “the” has been deleted.
(6) In line 101 of Page 3, “remarkably” has been revised by “remarkable”.
(7) In line 115 of Page 3, “that” has been revised by “than”.
(8) In line 117 of Page 3, “gradual” has been revised by “gradually”.
(9) In line 253 of Page 6, “was” has been revised by “were”.
(10) In line 258 of Page 6, “decreasing” has been revised by “decreased”.
(11) In line 275 of Page 7, “acquire” has been revised by “acquiring”.
(12) In line 373 of Page 10, “fluorinated” has been inserted after “further”..
(13) In line 433 of Page 12, “was” has been revised by “were”.
(14) In line 450 of Page 12, “gradually” has been revised by “gradual”.
(15) In line 452 of Page 13, “testing” has been revised by “test”.
(16) In line 456 of Page 13, “realized” has been deleted.
(17) In line 462 of Page 13, “which” has been deleted.

Reviewer 2 Report
This paper described some BT conjugated polymers for application for photovoltaic materials.
Basically, this paper is just a sequel using slightly different polymers along their previous paper (Polymers, 2019, 11, 12), and no problems could be found from this viewpoint.
Therefore, it should be accepted by Polymers almost as it is.
However, some minor mistakes should be corrected at the
proof stage. For example,
L 404: In In order to probe -> In order to probe
That's all.
Author Response
Reviewer #2
This paper described some BT conjugated polymers for application for photovoltaic materials.
Basically, this paper is just a sequel using slightly different polymers along their previous paper (Polymers, 2019, 11, 12), and no problems could be found from this viewpoint.
Therefore, it should be accepted by Polymers almost as it is.
However, some minor mistakes should be corrected at the proof stage. For example, L 404: In In order to probe -> In order to probe
Response: Thank you very much for carefully reviewing our manuscript giving the invaluable advice in order to improve it. We have carefully read revised the manuscript according to other reviewer’s comments. Furthermore, “In In order to probe” has revised by “In order to probe” in the revised manuscript.
